# Effect of Cavitation Water Jet Peening on Properties of AlCoCrFeNi High-Entropy Alloy Coating

**Rui Wu, Yongfei Yang \*** , **Weidong Shi, Yupeng Cao, Yu Liu and Jinchao Zhang \***

School of Mechanical Engineering, Nantong University, Nantong 226019, China; wr2109310037@163.com (R.W.)
\* Correspondence: yyf2020@ntu.edu.cn (Y.Y.); zhangjc@ntu.edu.cn (J.Z.)

**Abstract:** High-entropy alloys have been widely used in engineering manufacturing due to their hardness, good wear resistance, excellent corrosion resistance, and high-temperature oxidation resistance. However, it is inevitable that metallurgical defects, such as micro cracks and micro pores, are produced when preparing the coating, which affects the overall performance of the alloy to a certain extent. In view of this situation, cavitation water jet peening (CWJP) was used to strengthen the AlCoCrFeNi high-entropy alloy coating. The effect of CWJP impact time on the microstructure and mechanical properties of CWJP were investigated. The results show that CWJP can form an effective hardening layer on the surface layer of the AlCoCrFeNi high-entropy alloy. When the CWJP impact time was 4 h, the microhardness of the surface layer of the specimen was harder than that of 2 h and 6 h, and the CWJP impact time had little effect on the thickness of the hardening layer. Observing the surface of the untreated and CWJP-treated specimens using the EBSD test, it was evident that the microstructure was significantly homogenized, the grains were refined, and the proportion of small-angle grain boundaries increased. The system reveals the grain refinement mechanism of the AlCoCrFeNi high-entropy alloy coating during plastic deformation. This study aims to provide a new surface strengthening method for obtaining high-performance AlCoCrFeNi high-entropy alloy coatings.

**Keywords:** cavitation water jet peening; high-entropy alloy coating; microstructural evolution





## 1. Introduction

The revolutionary concept of high-entropy alloys (HEAs) was initially introduced by scholars, led by Junwei Ye et al. [1]. Utilizing the arc-melting method, they successfully synthesized HEAs with multiple principal elements. This breakthrough in alloy design addressed the limitations of traditional approaches, offering a wider array of element combinations in multi-principal element alloys, thereby expanding their application domains. Due to their exceptional properties, including high hardness, outstanding wear resistance, remarkable corrosion resistance, and resilience to high-temperature oxidation, HEAs have demonstrated significant potential and promising applications in critical engineering sectors, such as mechanical manufacturing, automotive and maritime industries, aerospace, and environmentally friendly processing [2,3].

In recent years, there has been a growing emphasis on researching high-entropy alloys (HEAs) due to their outstanding properties, such as high strength, hardness, and corrosion resistance. However, the cost of manufacturing HEAs is relatively high. Utilizing laser cladding technology on less expensive or less corrosion-resistant metal substrates to produce corrosion-resistant HEA coatings holds promising application prospects. Laser cladding technology, distinguished by its high laser beam power density, rapid heating and cooling cycles, minimal heat-affected zone and substrate deformation, versatile cladding powder selection, low dilution rate with the substrate resulting in a robust metallurgical bond, fine and uniform microstructure of the cladding layer, minimal macro and micro defects, and ease of automation [4–6], has garnered attention. This positions laser cladding

as an appealing method for fabricating the cladding layers of HEAs, ensuring not only the outstanding properties of HEA materials but also a secure bond between the cladding layer and the substrate, while minimizing the thermal impact on the substrate.

During the coating preparation, metallurgical defects, like microcracks and micropores, are inevitably generated, affecting the overall performance of the alloy. Shot peening technology [7,8] is one of the effective methods used to reduce fatigue and improve the lifespan of the components. In traditional shot peening treatment, high-speed projectiles are directed onto the surface of the components, inducing plastic deformation in the surface layer. This results in the formation of a reinforced layer with a certain thickness, and the reinforced layer develops higher residual stresses. When the component is subjected to loads, the compressive stresses on the surface of the component can counteract a portion of the applied stress, thereby enhancing the fatigue strength of the component. However, from another perspective, shot peening technology generates fine particles, such as dust, which can lead to serious environmental pollution. Additionally, when using shot peening for surface treatment, the impact force is significant, making it prone to deformation of the treated specimens. Therefore, traditional mechanical shot peening techniques are gradually proving insufficient to meet current processing requirements.

Over the past few decades, to overcome the limitations of traditional mechanical shot peening techniques, the development of Cavitation Water Jet Peening (CWJP) has emerged. The cavitation water jet, as an innovative jetting technology, offers advantages such as low cost, environmental friendliness, and high efficiency and safety. It has found wide applications in various industrial sectors, including mechanical processing, oil drilling, and microbiological degradation. Applying cavitation jetting to enhance the surface of metals is known as cavitation water jet peening technology [9,10]. In comparison to traditional shot peening, cavitation water jet peening involves no collisions between solid particles, resulting in a smoother surface after the strengthening treatment. Moreover, cavitation water jet peening introduces no pollution, aligning with the concept of sustainable and green industrial development.

The utilization of cavitation in high-pressure water jets was first proposed by R.E. Kohl [11], who initiated research into cavitation water jets. Since then, continuous studies on cavitation water jets have been conducted and widely applied to the surface treatments of materials. Pioneering contributions were made by Soyama et al., who explored the use of cavitation water jet peening to improve the performance of materials. They conducted peening experiments on stainless steel and aluminum alloy specimens, revealing a 20%−50% increase in fatigue strength on the peened surfaces compared to untreated surfaces [12–15]. Takakuwa et al. [16,17] investigated the enhancement treatment of 316 L austenitic stainless steel and studied its inhibitory effect on hydrogen-assisted fatigue crack propagation. They found that extending the treatment time could reduce the crack propagation rate of hydrogen-containing specimens by 75% compared to untreated specimens. Other researchers introduced residual stress through cavitation water jet peening on the surfaces of SCR420H steel gears, gear shafts, and high-speed steel forming tool roots, showing varying degrees of improvement in the fatigue life of these components [18–21]. Han et al. [22,23] studied the influence of cavitation water jet peening on the structure of SAE 1070 spring steel and found that it created a stable residual stress layer near the surface through X-ray diffraction experiments.

A conventional submerged cavitation water jet refers to a continuous jet that injects high-speed water into a static water environment, causing intense shear cavitation and generating numerous cavitation bubbles [24]. However, its operation is limited to underwater conditions, preventing the treatment of large-sized components that cannot be submerged. To address this limitation, researchers have devised an artificial submerged nozzle, incorporating a ring-shaped sleeve around the nozzle to create two water channels [25]. High-pressure water is injected through the inner nozzle, while low-pressure water with a larger flow rate is directed into the ring-shaped sleeve, creating a submerged environment for the inner nozzle. The high-speed water flow from the inner nozzle interacts with

the surrounding low-pressure water, inducing shear cavitation. Previous studies have mainly focused on the influence of cavitation water jet peening on the impact properties of traditional alloys, and there are relatively few studies on high-entropy alloy coatings.

In this paper, the AlCoCrFeNi high-entropy alloy coating was strengthened by cavitation water jet peening (CWJP), and the effects of CWJP on its microstructure and mechanical properties under different impact times were studied. The distribution of the microhardness of AlCoCrFeNi high-entropy alloy coatings along the depth direction was observed under different impact times. The interface response model of AlCoCrFeNi high-entropy alloy coatings impacted by cavitation jets was established, and the grain refinement mechanism of the AlCoCrFeNi high-entropy alloy coating during plastic deformation was systematically revealed. Therefore, investigating the mechanism behind the cavitation-water-jet-enhanced laser cladding of AlCoFeNi coatings is of significant importance in advancing the application of cavitation water jet peening technology.

## 2. Experimental Procedures

### 2.1. Materials and Methods

This study employed laser cladding technology to prepare the materials, using 304 stainless steel as the substrate The chemical composition of 304 stainless steel is detailed in Table 1. The cladding powder consisted of AlCoCrFeNi high-entropy alloy powder. Its specific chemical composition is provided in Table 2. The powder average particle size was 86.34 μm, and its scanning electron microscopy (SEM) image is presented in Figure 1. Laser cladding technology, a novel surface modification technique, utilizes a high-energy density laser beam as a heat source to melt and solidify the cladding material (powder, wire, or sheet) with the substrate surface. This process forms a metallurgical bond, significantly enhancing the wear resistance, heat resistance, and corrosion resistance of the substrate surface. A schematic diagram of the laser cladding equipment used in this study is presented in Figure 2. Laser cladding was performed with a cladding power of 2700 W, a cladding layer thickness of 1 mm, and a scanning speed of 16 mm/s along the cladding direction. After laser cladding, specimens measuring $40 \times 41 \times 18$ mm$^3$ were cut from the cladding area using wire EDM. The specimens were then ground with 80#, 320#, 600#, 800#, 1200#, and 1500# metallographic sandpaper, followed by polishing using a metallographic polishing machine until a mirror-like finish was achieved.

**Table 1.** Chemical composition of 304 stainless steel.

| Element | C | P | S | Mn | Cr | Ni | N | Si | Fe |
|---|---|---|---|---|---|---|---|---|---|
| Wt.% | 0.08 | 0.035 | 0.015 | 2 | <19.5 | 8 | <0.1 | 0.75 | Bal. |

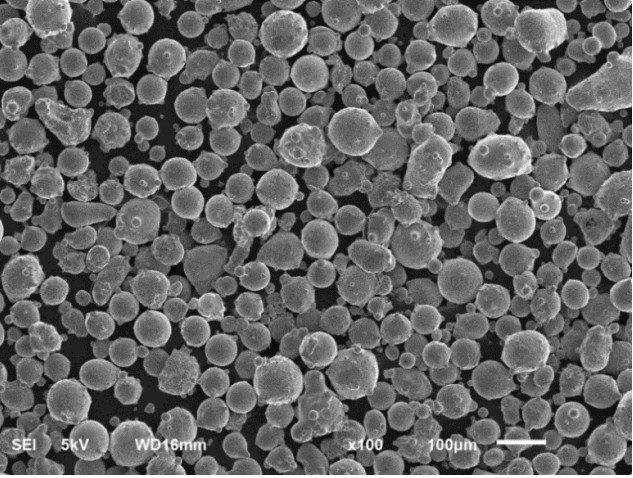

**Figure 1.** SEM image of AlCoCrFeNi high-entropy alloy powder.

**Table 2.** Chemical composition of high-entropy alloy powder.

| Element | Fe | Al | Co | Cr | Ni | O |
|---|---|---|---|---|---|---|
| Wt.% | 23.39 | 5.63 | 24.56 | 21.65 | 24.61 | Bal. |

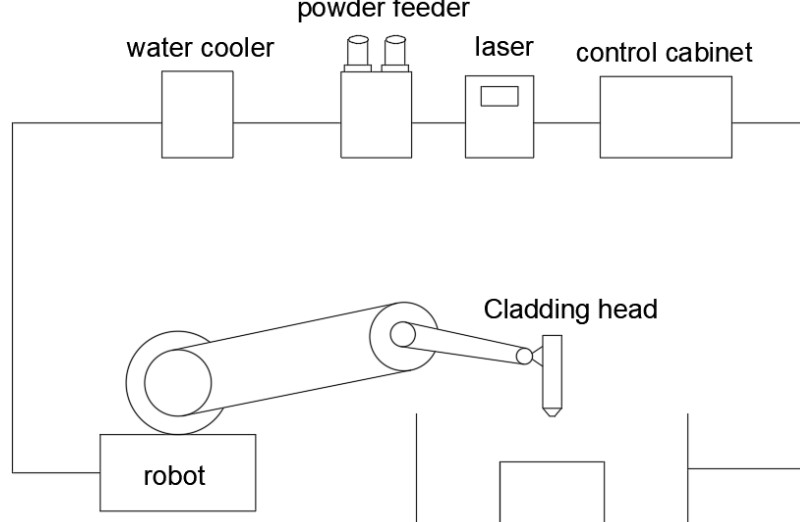

**Figure 2.** Image of laser cladding equipment.

Figure 3 illustrates the cavitation water jet peening (CWJP) apparatus used in the experiment. The power system includes all components necessary to supply sufficient flow to the nozzle, with both high-pressure (internal jet) and low-pressure (external jet) circuits. The high-pressure pipeline is equipped with a plunger pump unit with a power of 55 kW, reaching a maximum pressure of 2000 bar. It has a rated speed of 1485 r/min and a flow rate of 15 L/min. The high-pressure pipeline employs high-pressure hoses with specifications according to GB/T3683. The plunger pump, produced by Siemens, is driven by an AC variable frequency motor and is regulated by a frequency converter. The nozzle used in this experiment is cylindrical, and its physical appearance is depicted in Figure 4. The nozzle is securely fixed to a three-degrees-of-freedom movable stage. The low-pressure pipeline is equipped with a CHL20-3 centrifugal pump, with a power of 4 kW, a rated speed of 2900 r/min, and a flow rate of 20 m$^3$/h. During the experiment, water at a temperature of 25 °C served as the fluid medium. The fluid enters both the high-pressure and low-pressure pipelines from a water storage tank and is then sprayed by the internal and external nozzles, respectively.

In this study, AlCoCrFeNi high-entropy alloys were utilized to evaluate the erosive effects of the cavitation water jet by examining macroscopic and microscopic morphological changes in the material before and after impact. The specimens were securely held in a custom fixture on a horizontal platform. The platform was perpendicular to the axis of the nozzle, which was fixed on a horizontal guide rail using the fixture. The distance between the nozzle and the specimen could be adjusted by moving the guide rail through a screw mechanism. In this study, the experimental procedure initially involved laser cladding high-entropy alloy powder on a stainless steel substrate, and a total of 4 specimens were prepared using laser cladding technology. Subsequently, these specimens were polished, followed by CWJP treatment for 2, 4, and 6 h. Before the impact test, the specimen was cleaned in an ultrasonic cleaner for 20 min, followed by drying. The prepared specimen was then affixed to the fixture to commence the impact test. By assessing the surface changes in the AlCoCrFeNi high-entropy alloy after different impact durations, the study aimed to observe the evolution of cavitation-water-jet-induced cavitation erosion on the material's surface.

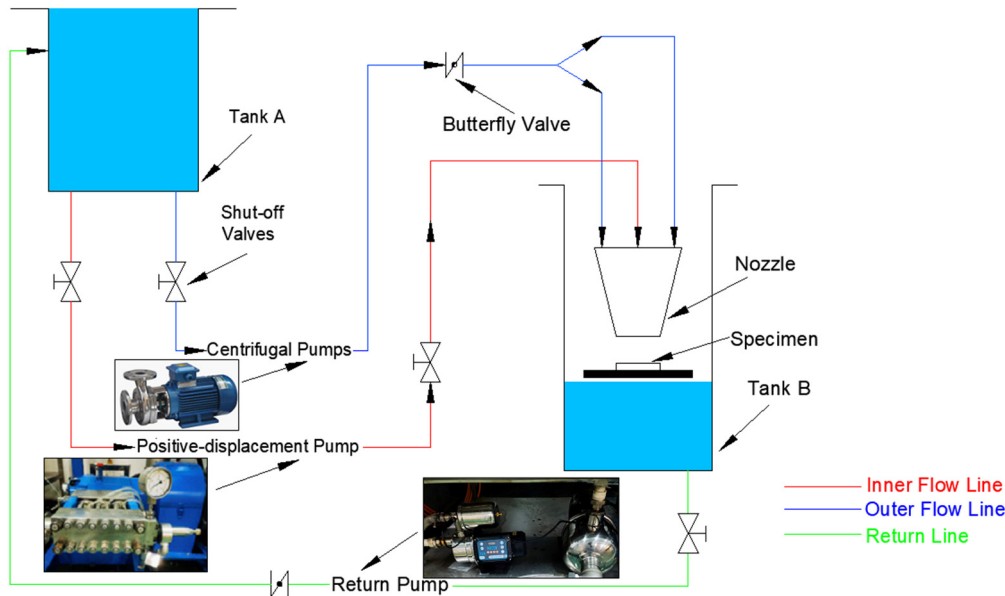

**Figure 3.** Device diagram of cavitation water jet peening (CWJP).

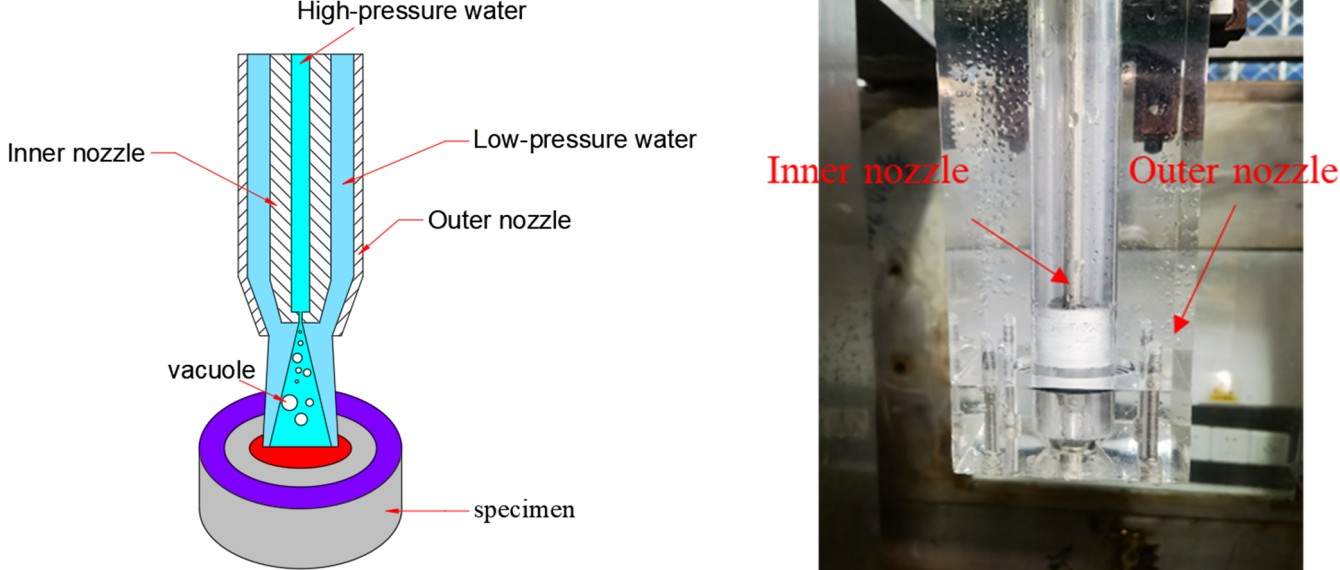

**Figure 4.** Schematic diagram of an artificially submerged cavitation water jet and a physical schematic diagram of the nozzle structure.

### 2.2. Microhardness Measurement

The microhardness distribution of all specimens was measured using the TWVS-1 digital micro Vickers hardness tester. Before obtaining the hardness measurement, the detection surface of the specimen underwent polishing and cleaning using an ultrasonic cleaner to enhance the accuracy of the hardness readings. The microhardness testing was carried out under a load of 0.981 N and for a duration of 15 s to obtain the specimen hardness distribution profiles. Microhardness measurements were conducted at unequal intervals, with a spacing of 50 μm between each indentation. Each microhardness value was obtained by averaging four measurements taken at the same depth.

### 2.3. Microstructure Characterization and Roughness Detection

Following cavitation water jet peening, the surface of the AlCoCrFeNi high-entropy alloy underwent plastic deformation. In this experiment, a NanoFocus Usurf confocal

microscope was used to observe the post-jetting surface, and its built-in post-processing software was employed for roughness extraction. Additionally, scanning electron microscopy (SEM) (Quanta 650F, FEI Corporation, Hillsboro, OR, USA) was utilized to observe the microstructure of the sample surfaces before and after treatment.

### 2.4. EBSD Detection

After CWJP treatment, specimens subjected to both untreated and 4 h CWJP treatment were selected for the EBSD analysis. The microstructural features of the specimens, including grain distribution, orientation errors, and texture, were observed using the Electron Backscatter Diffraction (EBSD) technique. In this study, prior to the EBSD analysis, the sample surfaces were chemically polished at 45–55 °C using a chemical polishing agent, followed by rinsing with water. Subsequently, the specimens were immersed in a room-temperature demounting solution (10% degreaser and 90% water) for 1–2 s to remove any residual film, followed by a final water rinse. After specimen preparation, the EBSD analysis was performed on the side of the sample surface located near the exposed aerodynamic jet.

### 3. Results and Discussion

#### 3.1. Microstructure Characterization and Roughness Analysis

Figure 5 presents the three-dimensional surface morphology of AlCoCrFeNi high-entropy alloy specimens at different impact durations, with corresponding surface roughness values provided in Table 3. Figure 5a illustrates the surface morphology of the specimens after laser cladding and polishing, but without cavitation water jet peening (CWJP) treatment. Due to polishing and grinding treatments, the surface appeared relatively smooth with only fine scratches, resulting in a roughness value of approximately 0.014 μm. In Figure 5b, the surface morphology of a shot-peened specimen with a 2-h impact duration is shown, and its surface roughness was measured at 0.851 μm. As the cavitation bubbles ruptured on the specimen surface, the high-pressure impacts from microjets and shock waves resulted in the formation of small-diameter, shallow plastic deformation pits on the specimen surface. There was a minimal amount of plastic deformation in the surface layer. Consequently, after a 2 h impact duration, the shot peening effect was relatively weak and not prominently evident.

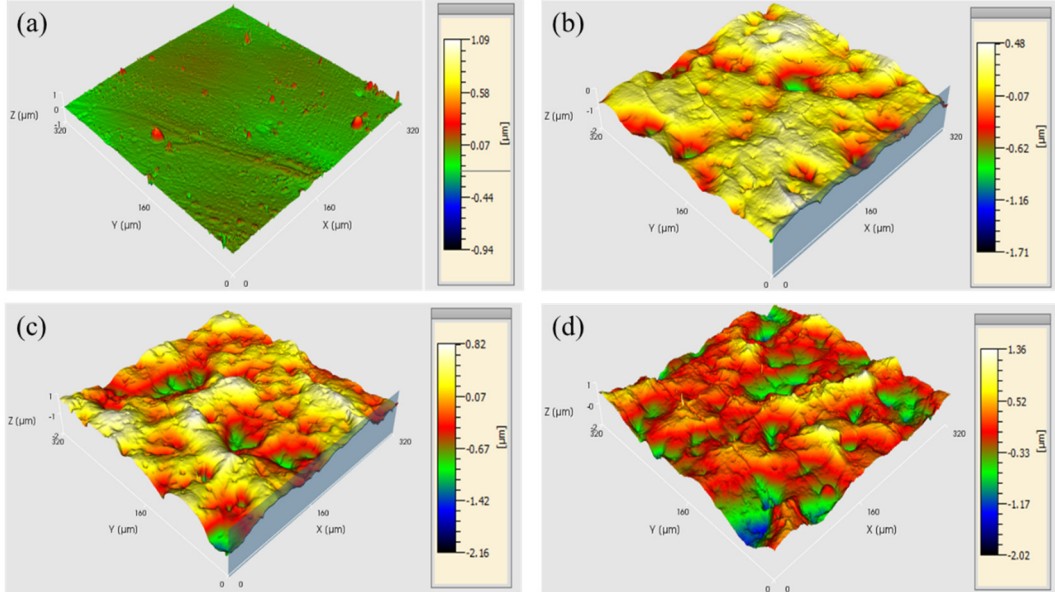

**Figure 5.** Three-dimensional morphology of the surface of high-entropy alloy specimens under different impact times. (**a**) without CWJP; (**b**) 2hCWJP; (**c**) 4hCWJP; (**d**) 6hCWJP.

**Table 3.** Surface roughness of specimens under different impact times.

| Specimen | unCWJP | 2hCWJP | 4hCWJP | 6hCWJP |
|----------|--------|--------|--------|--------|
| Ra/μm | 0.014 | 0.851 | 2.298 | 4.204 |

Figure 5c depicts the three-dimensional surface morphology of shot-peened specimens with a 4-h impact duration. With an increasing duration of cavitation bubble impact, compared to the 2-h impact specimens, the roughness value of the impact surface significantly rose to approximately 2.298 μm. The density of the pits generated by the impact increased, which was accompanied by an enlargement in both diameter and depth. When the impact duration reached 6 h, as shown in Figure 5d, the surface morphology of the shot-peened specimens continued to be exposed to the shock waves produced by cavitation bubble collapse. The roughness value rapidly increased to around 4.204 μm, and there was a significant increase in the depth of the pits generated by the impact. Additionally, the density of the pits increased with prolonged impact duration.

### 3.2. Microhardness Analysis

Figure 6 illustrates the variation in microhardness with depth for AlCoCrFeNi high-entropy alloy specimens before and after CWJP treatment. The microhardness distribution of the untreated specimens in the depth direction was relatively stable, with an average microhardness of 515.96 HV. After CWJP treatment, due to grain refinement and dislocation strengthening in the surface layer, the microhardness was highest at the surface. The CWJP impact time has a significant impact on the microhardness of the specimen surface, showing an increasing trend followed by a decrease with prolonged impact time.

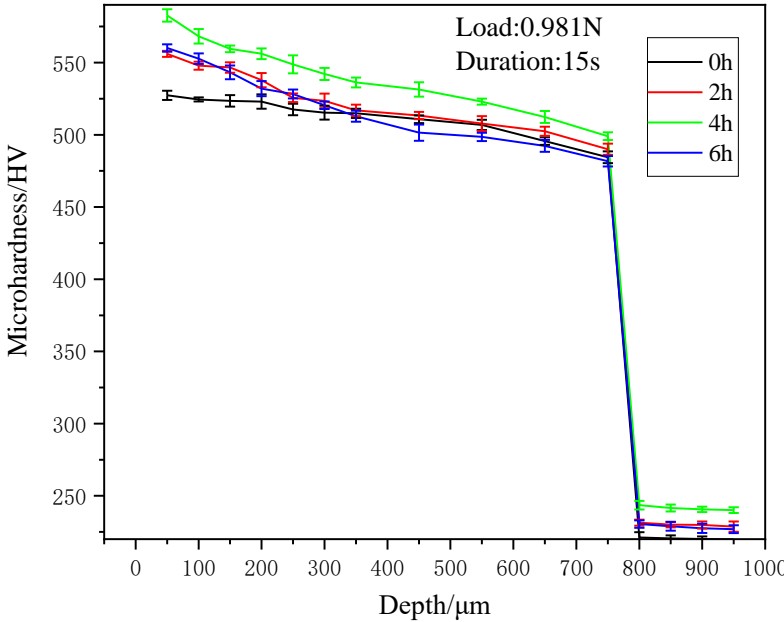

**Figure 6.** Distribution of microhardness with depth of the AlCoCrFeNi high-entropy alloy specimens.

When the CWJP impact time was 2 h, the highest microhardness at the specimen surface was 556.04 HV, indicating a noticeable improvement compared to the untreated specimens. At 4 h of CWJP impact time, the highest microhardness at the specimen surface reached 582.66 HV, and the rate of improvement began to slow. With a CWJP impact time of 6 h, the highest microhardness at the specimen surface was 560.12 HV, showing a more gradual increase. As the depth of the specimen increased, the microhardness gradually decreased until it stabilized at a hardness value identical to the base material. This suggests that, after CWJP strengthening, an effective hardened layer forms on the surface of the

AlCoCrFeNi high-entropy alloy. The CWJP impact time exhibits minimal influence on the thickness of the hardened layer, which measured approximately 780 μm.

### 3.3. Microstructure

To analyze the microstructural changes in AlCoCrFeNi high-entropy alloy specimens under different impact durations, four different specimens (untreated with cavitation water jet peening or treated for 2 h, 4 h, or 6 h with cavitation water jet peening) were selected for SEM analysis. Figure 7a represents the AlCoCrFeNi high-entropy alloy specimen untreated with cavitation water jet peening. The laser cladding process used to prepare the alloy coating induced metallurgical defects, such as microcracks and micropores, on its surface. In Figure 7b, the SEM morphology of the high-entropy alloy specimen subjected to a 2-h impact test is shown. Due to the relatively short impact duration, numerous cavities remained intact and did not collapse on the material surface, leading to an insufficient generation of extensive plastic deformation. The surface exhibited pitting corrosion after the 2-h impact test, but with a low density.

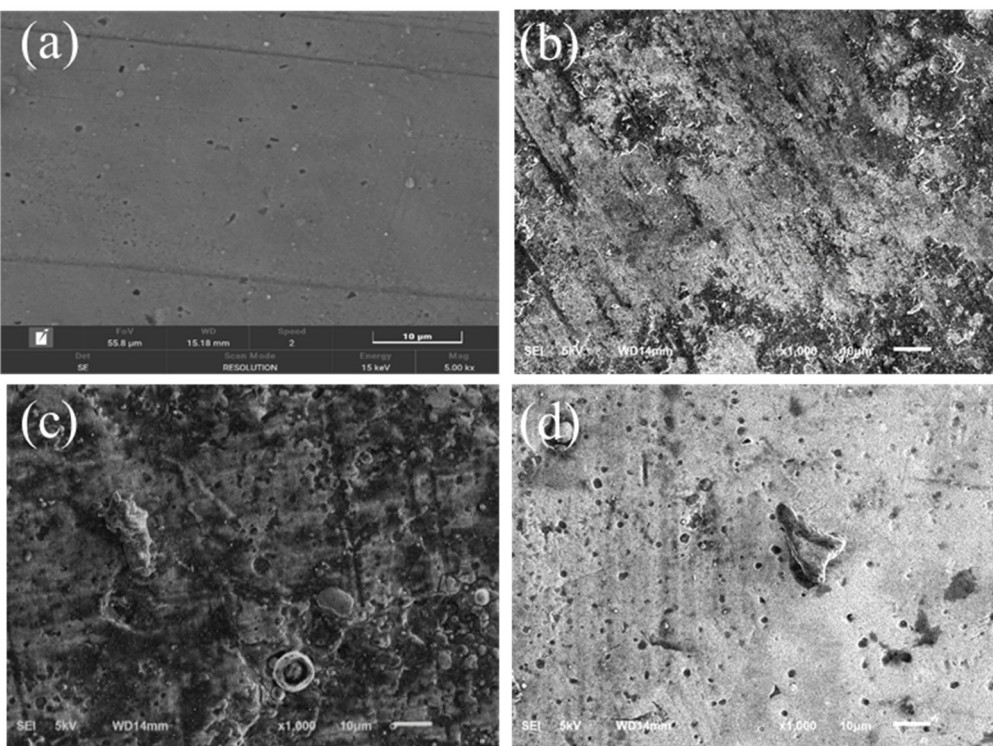

**Figure 7.** SEM morphology of AlCoCrFeNi high-entropy alloy specimens under different impact times. (**a**) unCWJP; (**b**) 2hCWJP; (**c**) 4hCWJP; (**d**) 6hCWJP.

As shown in Figure 7c, with an increase in the impact duration to 4 h, the surface plastic deformation of the specimen continued to grow. As the cavities collapsed on the specimen surface, smaller pitting corrosion features accumulated, overlapped, and spread to the surrounding area. This resulted in the appearance of larger diameter pits. In Figure 7d, with the cavitation water jet impact duration further extended to 6 h, the surface plastic deformation of the specimen continued to increase. The impact of cavities collapsing further enlarged the depressed regions on the specimen surface, and the number of pits significantly increased. Larger diameter pitting corrosion features also became more prevalent.

### 3.4. EBSD Haracterization Analysis

Figure 8 presents the column charts depicting the distribution of grain sizes in the high-entropy alloy before and after CWJP treatment. From the chart, it is evident that the

untreated high-entropy alloy exhibited a significant proportion of grains larger than 100 μm, accounting for approximately 26.6% of the total grain size. The grain size distribution was coarse, with less than 10% of grains smaller than 20 μm, indicating a bimodal distribution of grain sizes. After 4 h of CWJP treatment, there was an overall reduction in grain size. Grains larger than 100 μm now constitute only 17% of the total grain size. The bimodal distribution of grain sizes was alleviated, and the microstructure became notably more uniform with finer grains compared to the untreated state.

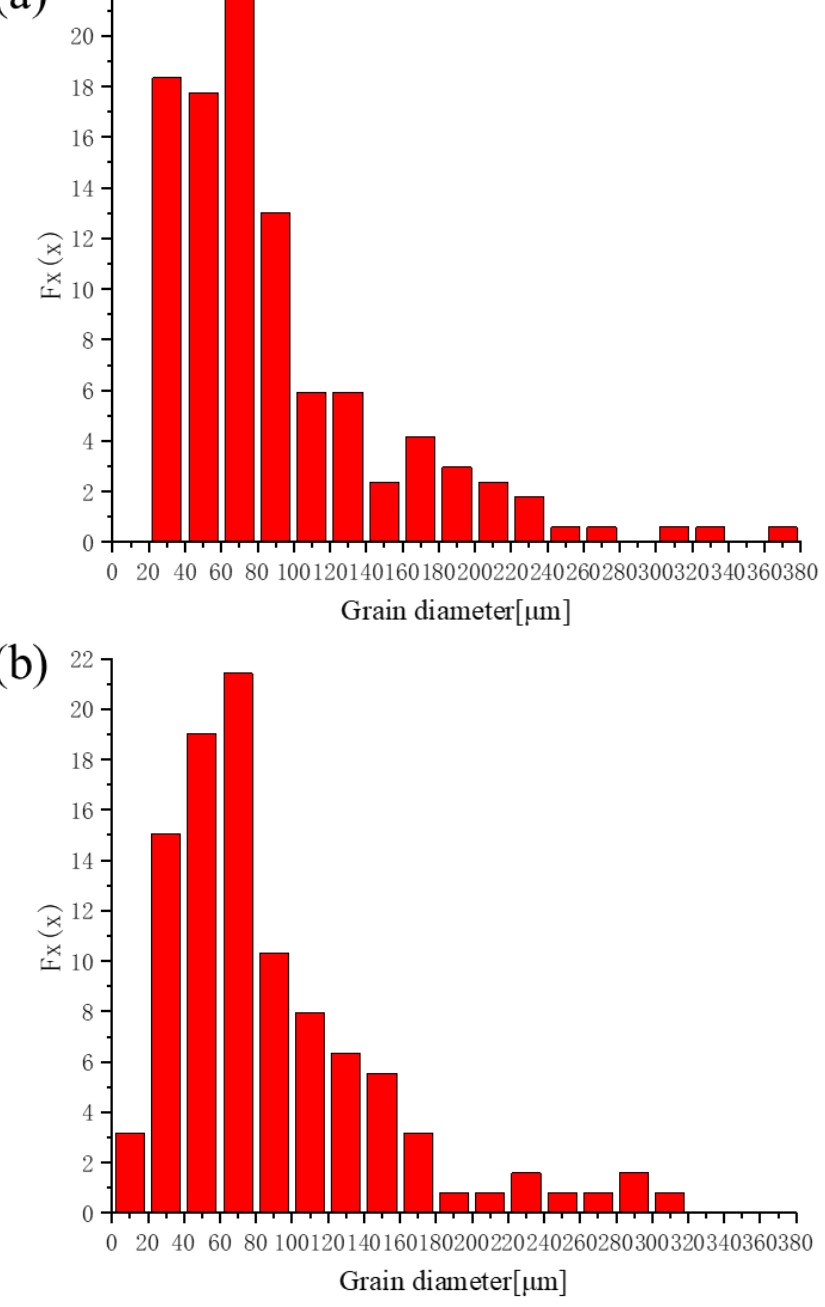

**Figure 8.** Histogram of the grain size distribution of the high-entropy alloy. (**a**) unCWJP; (**b**) 4hCWJP.

Table 4 provides the statistical data for the average grain size of specimens before and after 4 h of CWJP treatment. Analysis of Table 4 reveals that the original specimen exhibited non-uniform grain sizes, with the maximum short-axis length of grains reaching 436.42 μm, albeit in small quantities. The smallest grain measured only 13.203 μm, and the average grain size was relatively large at 85.737 μm.

**Table 4.** Statistical table of the average grain size of specimens before and after CWJP treatment.

| Grain Size | Max/μm | Min/μm | Mean/μm |
|---|---|---|---|
| unCWJP | 436.42 | 10.20 | 85.73 ± 67.74 |
| 4hCWJP | 310.69 | 13.20 | 67.04 ± 48.10 |

After CWJP treatment, there was a noticeable homogenization of grain sizes, and the larger grains in the material were significantly refined. The maximum grain size was reduced to 310.69 μm, and the average grain size decreased to 67.044 μm compared to the untreated material. This indicates that CWJP treatment effectively alleviates the problem of grain coarsening during the laser cladding process. Additionally, CWJP treatment induces the precipitation of finer grains, contributing to a reduction in the overall grain size distribution.

Figure 9a represents the distribution of misorientation angles in the high-entropy alloy specimens untreated with CWJP. In this context, misorientation angles between 2° and 15° are defined as low-angle grain boundaries, whereas those exceeding 15° are considered high-angle grain boundaries. As observed in Figure 9, both before and after CWJP treatment, the high-entropy alloy primarily exhibited high-angle grain boundaries. However, after CWJP treatment, there was a slightly higher proportion of high-angle grain boundaries. This is attributed to the deformation induced by CWJP treatment, which introduced a significant amount of small-angle grain boundaries. Some of these small-angle grain boundaries may disappear during subsequent dynamic recrystallization processes.

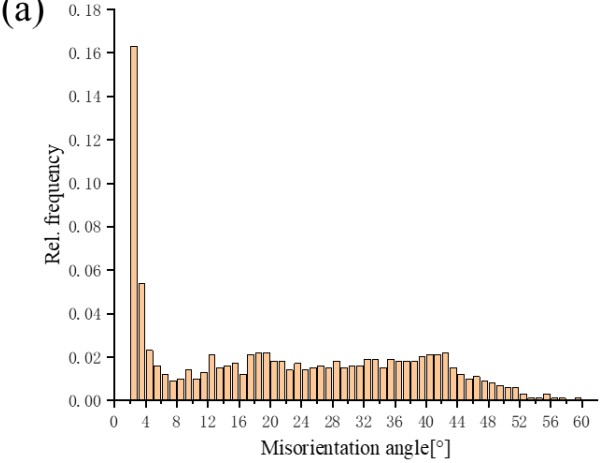

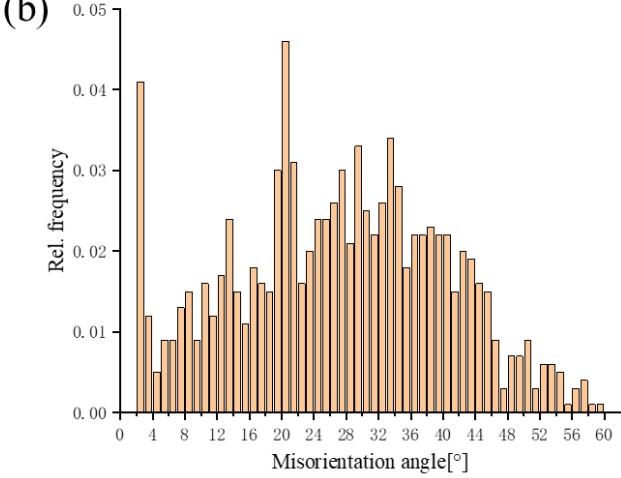

**Figure 9.** Distribution of orientation difference angles of high-entropy alloy specimens before and after CWJP treatment. (**a**) unCWJP; (**b**) 4hCWJP.

Figure 10 depicts the Electron Backscatter Diffraction (EBSD) grain orientation maps for the high-entropy alloy specimens untreated and treated with 4 h of CWJP.

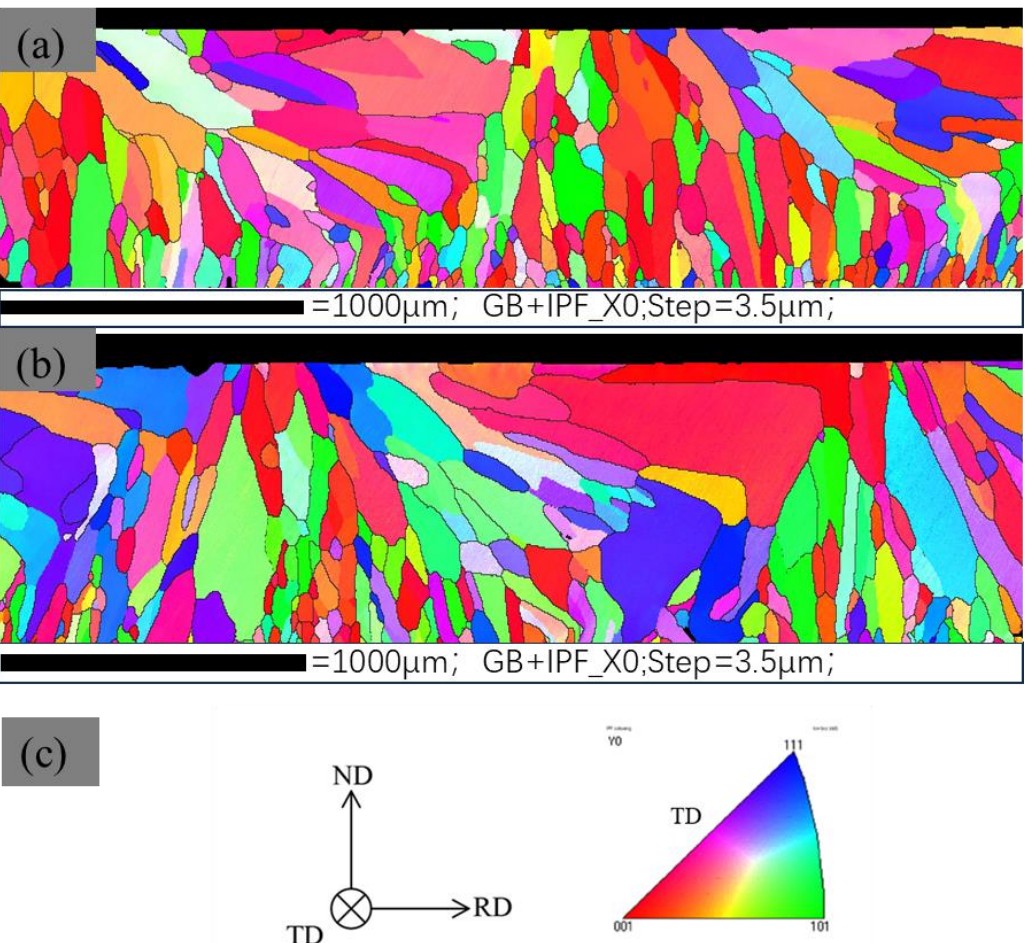

**Figure 10.** EBSD grain orientation map and inverse pole figure (IPF) of high-entropy alloy specimens before and after CWJP treatment. (**a**) unCWJP; (**b**) 4hCWJP; (**c**) inverse pole figure.

Figure 10a shows the grains of high entropy alloy specimens without CWJP treatment. The largest grains have a size of approximately 400 μm, whereas the smallest are around 20 μm. Figure 11 depicts the pole figures of high-entropy alloy specimens before and after CWJP treatment. When combined with the inverse pole figure, the orientation distribution law can be seen intuitively. The orientation distribution within the selected area of the material predominantly aligns along the (001) and (111) directions.

After 4 h of CWJP treatment, the high-entropy alloy specimens underwent dynamic recrystallization in the impacted region, resulting in equiaxed grains. After CWJP treatment, the shock wave deformed the metal. As the atomic diffusion ability increased, the elongated and broken grains of the deformed metal transformed into new uniform and fine equiaxed grains through secondary nucleation and growth. The orientation distribution of the selected area of the material was mostly concentrated in the (001) direction, and the average grain size was 67 μm. Compared to the untreated alloy, the volume fraction of low-angle grain boundaries increased to 37%, whereas that of high-angle grain boundaries decreased to 63%, as shown in Figure 11. From Figure 10b, it is evident that during the cavitation water jet impact, the grains undergo deformation and bending due to the strengthening effect of cavitation. Dislocation accumulation and recombination take place within the elongated grains. In the process of plastic deformation, some sub-grains within the elongated grains rotate and recrystallize. During partial dynamic recrystallization,

some low-angle grain boundaries continuously absorb dislocations, increasing their angles and eventually transforming into high-angle grain boundaries.

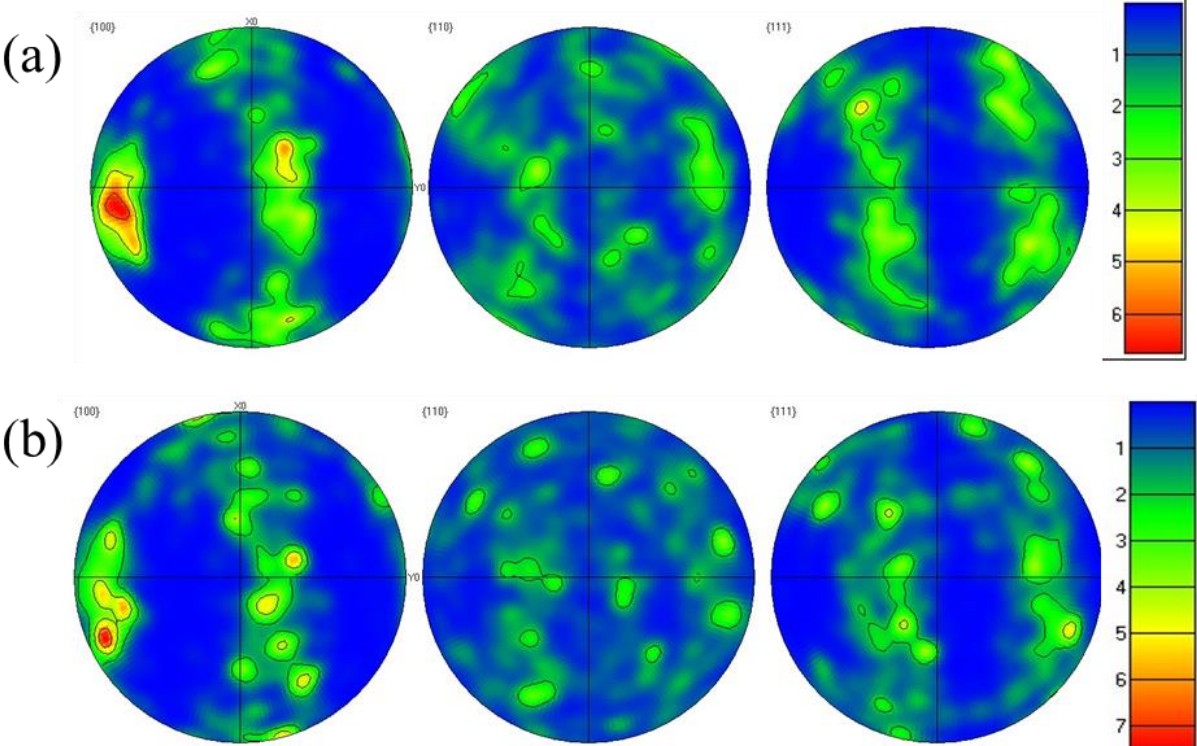

**Figure 11.** Pole figures of high-entropy alloy specimens before and after CWJP treatment. (**a**) unCWJP; (**b**) 4hCWJP.

The analysis of the EBSD results suggests that the impact of cavitation water jet shock waves induces the generation of significant residual stresses on the surface of the AlCoCr-FeNi high-entropy alloy. This stimulation leads to the formation of numerous dislocations, and the interaction of these dislocations with grain boundaries in the AlCoCrFeNi high-entropy alloy, along with their self-entanglement, significantly enhances the plasticity of the alloy. Furthermore, after the cavitation water jet impact, precipitated phases act as anchors for dislocations, hindering the further growth of material grains, thereby refining the grain structure. Simultaneously, the entanglement between precipitated phases and dislocations greatly improves the mechanical properties of the AlCoCrFeNi high-entropy alloy. This microstructural adjustment at the microscopic level provides robust support for enhancing the plasticity and mechanical performance of the alloy.

### 3.5. Grain Refinement Mechanism Induced by CWJP

By analyzing the grain size, grain orientation, and orientation difference angle of the cladding layer interface, the interface response model of cavitation jet impacts on the AlCoCrFeNi high-entropy alloy coating was established, as shown in Figure 12. After the laser cladding was completed, due to the large temperature gradient between the cladding layer and the body, the grain nucleation and solidification rates were high. The molten metal in the molten pool re-nucleated with trace elements as the core, and relatively fine grains were generated on the surface. After 4 h of CWJP treatment, a large proportion of the shock wave generated by cavitation was transmitted into the cladding layer, resulting in the refinement of surface grains and the formation of an effective hardened layer (Figure 13).

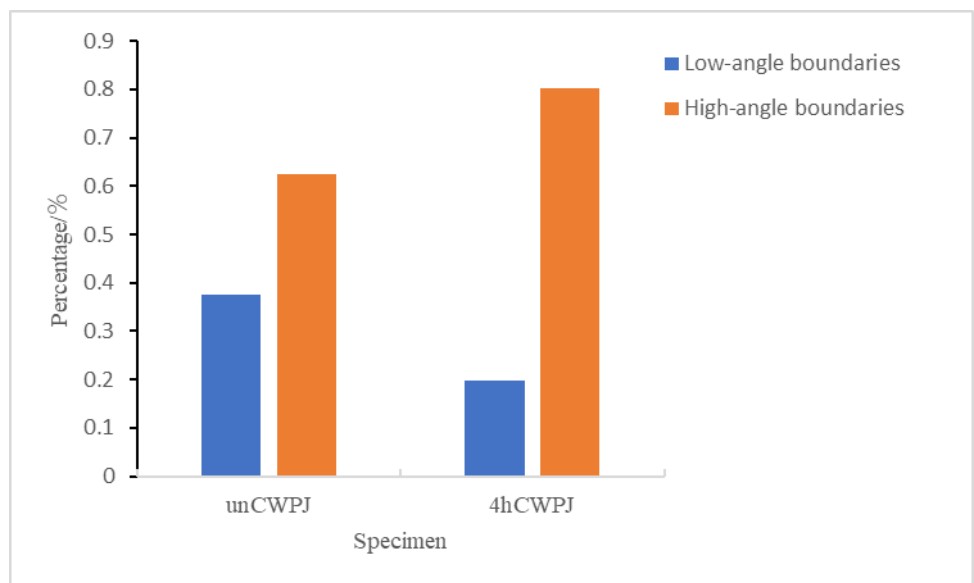

**Figure 12.** Low-angle and high-angle grain boundary components of specimens before and after CWJP treatment.

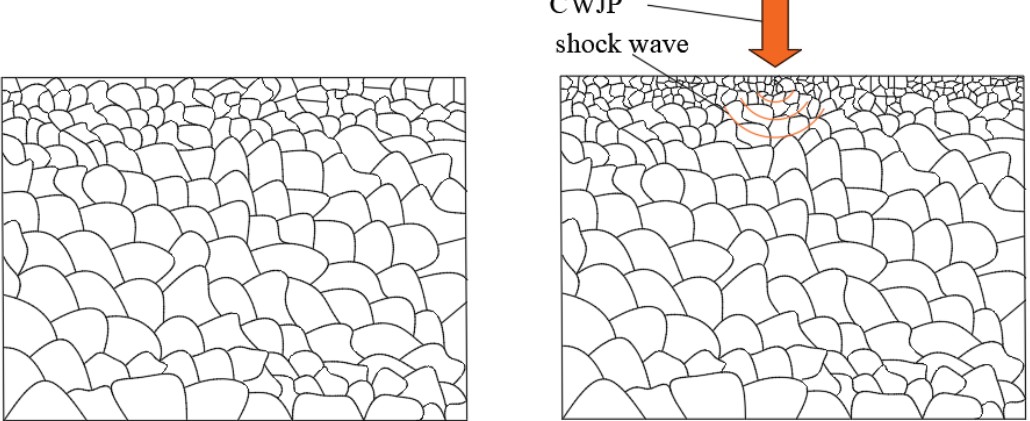

**Figure 13.** Interface response model of CWJP impacting the AlCoCrFeNi high-entropy alloy coating.

### 4. Conclusions

This study employed cavitation water jet peening (CWJP) to enhance the surface of the AlCoCrFeNi high-entropy alloy and investigated the impact of CWJP on its microstructure and mechanical properties under different impact durations. Changes in the surface morphology and roughness of the AlCoCrFeNi high-entropy alloy under various impact durations were observed. The microhardness distribution along the depth was measured for different CWJP impact durations. An EBSD (Electron Backscatter Diffraction) analysis was conducted on the surface of the AlCoCrFeNi high-entropy alloy before and after impact. The reasons for grain refinement during the plastic deformation process of the AlCoCrFeNi high-entropy alloy were systematically analyzed. The specific conclusions are summarized as follows:

(1)  CWJP treatment induces plastic deformation on the surface of the AlCoCrFeNi high-entropy alloy. With increasing impact time, the specimen surface experiences prolonged exposure to shock waves, resulting in a gradual increase in surface roughness. The depth of pits and the density of pit distribution also increase.

(2)  After CWJP strengthening, an effective hardened layer can form on the surface of the AlCoCrFeNi high-entropy alloy. The CWJP impact time has minimal influence on the thickness of the hardened layer, which is approximately 780 μm. When the CWJP

impact time is 4 h, the microhardness of the specimen surface surpasses that of 2 h and 6 h of impact time. This is attributed to insufficient impact intensity with shorter peening times, whereas excessively long peening times lead to erosion of the initially formed strengthened layer on the material surface.

(3)  In the process of preparing AlCoCrFeNi high-entropy alloy coatings using laser cladding, inevitable metallurgical defects, such as microcracks and microvoids, occur on the surface. As the impact time increases, surface plastic deformation continues to grow. There is a trend of pit accumulation, overlapping, and spreading, resulting in an enlargement of the depressed areas caused by the impact of collapsing cavitation bubbles. The number of pits significantly increases, and larger-diameter pits also become more prevalent.

(4)  After CWJP treatment, an overall reduction in grain size occurs, alleviating the bimodal distribution of grain sizes. The microstructure becomes notably more uniform, and the grains become finer. After CWJP treatment, there is an increase in the proportion of low-angle grain boundaries, indicating that the treatment induces significant deformation in the high-entropy alloy material, introducing a substantial amount of low-angle grain boundaries. Due to the incomplete nature of dynamic recrystallization, there are still low-angle grain boundaries that have not transformed into high-angle grain boundaries.

**Author Contributions:** Conceptualization, W.S.; Methodology, R.W. and Y.C.; Validation, Y.L.; Investigation, R.W., Y.C. and J.Z.; Resources, Y.Y.; Writing—original draft, R.W.; Writing—review & editing, Y.L. and J.Z.; Supervision, Y.Y.; Project administration, W.S. All authors have read and agreed to the published version of the manuscript.

**Funding:** This research was funded by the National Key Research and Development Project of China (No. 2019YFB 2005300), the National High-Tech Ship Scientific Research Project of China (No. MIIT [2019] 360), the National Natural Science Foundation of China (No. 51979138), the National Natural Science Foundation of China (No. 273746), the National Natural Science Foundation of China (No. 51979240), Jiangsu Natural Science Research Project (No. 19KJB470029), China Postdoctoral Science Foundation (No. 273746), and the Natural Science Foundation of Jiangsu Province (No. BK20220609). The project was funded by China Postdoctoral Science Foundation (No. 2022TQ0127), Open Research Subject of Key Laboratory of Fluid Machinery and Engineering (Xihua University) of China (No. LTDL-2022001).

**Institutional Review Board Statement:** Not applicable.

**Informed Consent Statement:** Not applicable.

**Data Availability Statement:** Data are contained within the article.

**Conflicts of Interest:** The authors declare no conflict of interest.

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
