# Peer review of "Effect of Cavitation Water Jet Peening on Properties of AlCoCrFeNi High-Entropy Alloy Coating"

_coatings, doi:10.3390/coatings13111972_

Round 1

Reviewer 1 Report

Comments and Suggestions for Authors

I have reviewed the manuscript titled “Effect of Cavitation Water Jet Peening on Properties of AlCoCrFeNi High-Entropy Alloy Coating” submitted to Coatings by the authors. The manuscript presents a study on the effect of cavitation water jet peening (CWJP) on the microstructure and mechanical properties of a high-entropy alloy coating obtained by laser cladding. The manuscript reports the changes in surface morphology, roughness, microhardness, grain size, and grain boundaries of the coating under different CWJP impact times.

I appreciate the efforts made by the authors to conduct this study and to write this manuscript. However, I regret to inform you that I cannot recommend the publication of this manuscript in Coatings in its current form. The manuscript has several major flaws and weaknesses that need to be addressed before it can be considered for publication. These are:

The manuscript lacks a clear objective, hypothesis, or research question. The manuscript does not explain why it is important or challenging to apply CWJP to a high-entropy alloy coating, or what the study aims to answer or achieve.

The manuscript lacks a critical discussion and interpretation of the results. The manuscript does not compare the results with those obtained by other techniques or materials, or explain the underlying mechanisms or phenomena behind the observed changes in the coating properties.

The manuscript lacks a clear contribution or implication of the study. The manuscript does not state what the impact or novelty of the study is, or how it advances the knowledge or application of CWJP or high-entropy alloy coatings.

Therefore, I suggest that the authors revise the manuscript thoroughly and address the above-mentioned issues before resubmitting it to Coatings. I hope that the authors will take my comments and suggestions into consideration and improve the quality and clarity of their manuscript.

Author Response

Question1: The manuscript lacks a clear objective, hypothesis, or research question. The manuscript does not explain why it is important or challenging to apply CWJP to a high-entropy alloy coating, or what the study aims to reply or achieve.

Reply: Thanks for the suggestion. According to the expert 's suggestion, the abstract and introduction have been modified, the research purpose of this manuscript has been added, and the significance of CWPJ applied to high entropy alloy coating has been added.

Abstract: High entropy alloy has been widely used in engineering manufacturing due to its high hardness, good wear resistance, excellent corrosion resistance and high temperature oxidation resistance. However, it is inevitable to produce metallurgical defects such as micro cracks and micro pores when preparing the coating, which affects the overall performance of the alloy to a certain extent. In view of this situation, cavitation water jet peening (CWJP) was used to strengthen the AlCoCrFeNi high-entropy alloy coating. The effect of impact times of CWJP on its microstructure and mechanical properties were investigated. The results show that an effective hardening layer can be formed on the surface layer of AlCoCrFeNi high-entropy alloy after CWJP. When the CWJP impact time is 4 hours, the microhardness of the surface layer of the specimen is better than that of 2 hours and 6 hours, and the CWJP impact time has little effect on the thickness of the hardening layer. Observing the surface of the un-treated and CWJP treated specimens by EBSD test, it is found that the microstructure is significantly homogenized, the grains are refined, and the proportion of small angle grain boundaries increases. The system reveals the grain refinement mechanism of AlCoCrFeNi high-entropy alloy coating during plastic deformation. This study aims to provide a new surface strengthening method for obtaining high-performance AlCoCrFeNi high-entropy alloy coatings.

1.Introduction

The revolutionary concept of high-entropy alloys (HEAs) was initially introduced by scholars, led by Junwei Ye et al.[1]. Utilizing the arc-melting method, they successfully synthesized HEAs with multiple principal elements. This breakthrough in alloy design addressed the limitations of traditional approaches, offering a wider array of element combinations in multi-principal element alloys and consequently expanding their application domains. Due to their exceptional properties, including high hardness, outstanding wear resistance, remarkable corrosion resistance, and resilience to high-temperature oxidation, HEAs have demonstrated significant potential and promising applications in critical engineering sectors such as mechanical manufacturing, automotive and maritime industries, aerospace, and environmentally friendly processing[2-3].

In recent years, there has been an increasing focus on the research of high-entropy alloys (HEAs) due to their outstanding properties such as high strength, hardness, and corrosion resistance. However, the cost of manufacturing HEAs is relatively high. Utilizing laser cladding technology on less expensive or less corrosion-resistant metal substrates to produce corrosion-resistant HEA coatings holds promising application prospects. Laser cladding technology, distinguished by its high laser beam power density, rapid heating and cooling cycles, minimal heat-affected zone and substrate deformation, versatile cladding powder selection, low dilution rate with the substrate resulting in a robust metallurgical bond, fine and uniform microstructure of the cladding layer, minimal macro and micro defects, and ease of automation [4-6], has garnered attention. This positions laser cladding as an appealing method for fabricating cladding layers of HEAs, ensuring not only the outstanding properties of HEA materials but also a secure bond between the cladding layer and the substrate while minimizing the thermal impact on the substrate.

During the coating preparation, metallurgical defects like microcracks and micropores are inevitably generated, affecting the overall performance of the alloy.Shot peening technology[7-8] is one of the effective methods to reduce fatigue and improve the lifespan of components. In traditional shot peening treatment, high-speed projectiles are directed onto the surface of components, inducing plastic deformation in the surface layer. This results in the formation of a reinforced layer with a certain thickness, and the reinforced layer develops higher residual stresses. When the component is subjected to loads, the compressive stresses on the surface of the component can counteract a portion of the applied stress, thereby enhancing the fatigue strength of the component.However, from another perspective, shot peening technology generates fine particles, such as dust, which can lead to serious environmental pollution. Additionally, when using shot peening for surface treatment, the impact force is significant, making it prone to deformation of the treated specimens. Therefore, traditional mechanical shot peening techniques are gradually becoming inadequate to meet current processing requirements.

Over the past few decades, to overcome the limitations of traditional mechanical shot peening techniques, the development of Cavitation Water Jet Peening (CWJP) has emerged. Cavitation water jet, as an innovative jetting technology, offers advantages such as low cost, environmental friendliness, and high efficiency and safety. It has found wide applications in various industrial sectors, including mechanical processing, oil drilling, and microbiological degradation. Applying cavitation jetting to enhance the surface of metals is known as cavitation water jet peening technology[9-10]. In comparison to traditional shot peening, cavitation water jet peening involves no collisions between solid particles, resulting in a smoother surface after the strengthening treatment. Moreover, cavitation water jet peening introduces no pollution, aligning with the concept of sustainable and green industrial development.

Conventional submerged cavitation water jet refers to a continuous jet that injects high-speed water into a static water environment, causing intense shear cavitation and generating numerous cavitation bubbles[24]. However, its operation is limited to underwater conditions, preventing the treatment of large-sized components that cannot be submerged. To address this limitation, researchers have devised an artificial submerged nozzle, incorporating a ring-shaped sleeve around the nozzle to create two water channels[25]. High-pressure water is injected through the inner nozzle, while low-pressure water with a larger flow rate is directed into the ring-shaped sleeve, creating a submerged environment for the inner nozzle. The high-speed water flow from the inner nozzle shears with the surrounding low-pressure water, inducing shear cavitation. Previous studies have mainly focused on the influence of cavitation water jet peening on the impact properties of traditional alloys, and there are relatively few studies on high-entropy alloy coatings.

Question2: The manuscript lacks a critical discussion and interpretation of the results. The manuscript does not compare the results with those obtained by other techniques or materials, or explain the underlying mechanisms or phenomena behind the observed changes in the coating properties.

Reply: Thanks for the suggestion. According to the expert 's suggestion, the potential mechanism behind the observed coating performance changes has been added to the text and marked in red. 

3.5 Grain refinement mechanism induced by CWJP

By analyzing the grain size, grain orientation and orientation difference angle of the cladding layer interface, the interface response model of cavitation jet impacting AlCoCrFeNi high-entropy alloy coating is established as shown in Figure 12. After the laser cladding is completed, due to the large temperature gradient between the cladding layer and the body, the grain nucleation rate and solidification rate are high. The molten metal in the molten pool re-nucleates with trace elements as the core, and relatively fine grains are generated on the surface. After 4hCWJP treatment, the shock wave generated by cavitation has a large proportion of transmission in the cladding layer, resulting in the refinement of the surface grains and the formation of an effective hardened layer.

Fig.12 Interface response model of CWJP impacting AlCoCrFeNi high-entropy alloy coating.

Question3: The manuscript lacks a clear contribution or implication of the study. The manuscript does not state what the impact or novelty of the study is, or how it advances the knowledge or application of CWJP or high-entropy alloy coatings.

Reply : Thanks for the suggestion. According to the expert 's advice, a description of the impact of the study has been added to the manuscript. In this paper, the AlCoCrFeNi high-entropy alloy coating was strengthened by cavitation water jet peening( CWJP ), and the effects of CWJP on its microstructure and mechanical properties under different impact time were studied. The distribution of microhardness of AlCoCrFeNi high entropy alloy coating along the depth direction under different impact time was observed. The interface response model of AlCoCrFeNi high-entropy alloy coating impacted by cavitation jet was established, and the grain refinement mechanism of AlCoCrFeNi high-entropy alloy coating during plastic deformation was systematically revealed.Therefore, the study of the mechanism behind cavitation water jet-enhanced laser cladding of AlCoFeNi coatings is of significant importance in advancing the application of cavitation water jet peening technology.

Reviewer 2 Report

Comments and Suggestions for Authors

Overview and general recommendation:

This is an interesting study of an advanced surface treatment methodology and its effect on hardness and microstructure. There are a few points of confusion where results and methods could be explained better, and a few points where scientific rigor could be improved. I think this manuscript can be published after the authors address my concerns below.

 Major comments:

1) Introduction: Most of the introduction section describes the state of the art well with appropriate citations, but there are several points where this is somewhat lacking and a better description of prior important work would improve the significance of the current work.

On page 2, line 64, the authors refer to work by Junwei Ye et al., but do not include a citation to this work. In fact, there are no citations in that entire paragraph about high entropy alloys.

On page 2, lines 81-90, the authors describe the cavitation water jet technology, but the description is confusing, and again, does not include any citations to sources where this is better described. The authors should add appropriate citations to seminal work in this area.

The final paragraph of the introduction (lines 91-102) contains no citations to back up the claims made. Also, the transition to the objective of the current work could be clarified.

 2) Experimental Procedures: There are many details of the different work performed in this section, but it was not clear what the performed experiment was. By reading the other sections, I am assuming the order of procedures was: 1) laser cladding of the high entropy powder particles on the stainless steel substrate, 2) polishing, 3) CWJP for different times (2, 4, or 6 hours), and 4) hardness and microstructure testing to identify differences between the different processed samples. Is this correct? Please revisit this section to clarify the details such as what was different between the samples and how many replicates were performed for each treatment.

3) Mechanical properties: The authors discuss how the proposed method has a positive impact on mechanical properties (including wear, fatigue, and corrosion resistance- see line 99) of high entropy alloys, but the only property measured in this study is the microhardness. The results would be significantly enhanced if another mechanical property could be measured as well.

4) Microhardness: There is a significant decrease in hardness between points at ~780 um deep and ~980 um deep, according to Figure 6. Were any depths between those points measured? It would be interesting to see a more exact point of where the hardened layer ends, which could potentially be identified with more tests in that crucial area, rather than leaving a section of 200 um between measurements. Also, the authors could consider adding error bars to the measurements in Fig. 6.

5) Table 4 lists the max, min, and mean values of grain size. Adding a standard deviation or other metric of dispersion of the measurements would help support the claim on line 274 that “the original specimen exhibits non-uniform grain sizes.”

Minor comments:

6) Abstract: The authors state on line 11 of the abstract that an impact time of 4 hours results in “better” microhardness than 2 or 6 hours. By “better,” do the authors mean “harder?”

7) Figure 2: The text on lines 113-114 say Fig. 2 will be a schematic diagram, but the actual figure is a photograph of lab equipment, without any labels or information that informs how the process works. An actual schematic diagram similar to Fig. 3, would be helpful to understand the laser cladding process.

8) Lines 187-188 describes a “shot-peened specimen without Cavitation Water Jet Peening (CWJP).” How can the specimen be shot-peened without the CWJP treatment? Do the authors mean a laser-clad and polished sample without CWJP? Or did they perform conventional shot-peening with some other method? If so, please describe that method.

9) The font size used in the axes and colorbars for Figures 5, 8, and 9 are too small to read. Please increase the font size. Also, Figure 6 does not have any axes labels or units. Please correct.

10) On line 306, the authors state that “the untreated high-entropy alloy exhibits elongated equiaxed grains…” How can a grain be both elongated and equiaxed, since the two descriptions are opposites?

11) Conclusion 2: On line 353, the authors state that the thickness of the hardened layer is approximately 650 um, but on line 231 and Fig. 6 of the text, the hardened layer thickness is given as 780 um. Please clarify.

Author Response

Question 1.1: On page 2, line 64, the authors refer to work by Junwei Ye et al., but do not include a citation to this work. In fact, there are no citations in that entire paragraph about high entropy alloys.

Reply: Thanks for the suggestion. According to the expert 's suggestion, relevant references have been inserted in the introduction,and insert reference to the high-entropy alloy in this section. The revolutionary concept of high-entropy alloys (HEAs) was initially introduced by scholars, led by Junwei Ye et al.[1]. Utilizing the arc-melting method, they successfully synthesized HEAs with multiple principal elements. This breakthrough in alloy design addressed the limitations of traditional approaches, offering a wider array of element combinations in multi-principal element alloys and consequently expanding their application domains. Due to their exceptional properties, including high hardness, outstanding wear resistance, remarkable corrosion resistance, and resilience to high-temperature oxidation, HEAs have demonstrated significant potential and promising applications in critical engineering sectors such as mechanical manufacturing, automotive and maritime industries, aerospace, and environmentally friendly processing[2-3].It has been marked in red font.

Question 1.2:On page 2, lines 81-90, the authors describe the cavitation water jet technology, but the description is confusing, and again, does not include any citations to sources where this is better described. The authors should add appropriate citations to seminal work in this area.

Reply: Thanks for the suggestion. According to the expert 's suggestion, a reference to the cavitating water jet technology has been inserted in this section (lines 81-90). Conventional submerged cavitation water jet refers to a continuous jet that injects high-speed water into a static water environment, causing intense shear cavitation and generating numerous cavitation bubbles[24]. However, its operation is limited to underwater conditions, preventing the treatment of large-sized components that cannot be submerged. To address this limitation, researchers have devised an artificial submerged nozzle, incorporating a ring-shaped sleeve around the nozzle to create two water channels[25]. High-pressure water is injected through the inner nozzle, while low-pressure water with a larger flow rate is directed into the ring-shaped sleeve, creating a submerged environment for the inner nozzle. The high-speed water flow from the inner nozzle shears with the surrounding low-pressure water, inducing shear cavitation. Previous studies have mainly focused on the influence of cavitation water jet peening on the impact properties of traditional alloys, and there are relatively few studies on high-entropy alloy coatings.It has been marked in red font.

Question 1.3:The final paragraph of the introduction (lines 91-102) contains no citations to back up the claims made. Also, the transition to the objective of the current work could be clarified.

Reply: Thanks for the suggestion. According to the expert 's suggestion, relevant references have been inserted in this paragraph ( lines 91-102 ). In this paper, the AlCoCrFeNi high-entropy alloy coating was strengthened by cavitation water jet peening( CWJP ), and the effects of CWJP on its microstructure and mechanical properties under different impact time were studied. The distribution of microhardness of AlCoCrFeNi high entropy alloy coating along the depth direction under different impact time was observed. The interface Reply model of AlCoCrFeNi high-entropy alloy coating impacted by cavitation jet was established, and the grain refinement mechanism of AlCoCrFeNi high-entropy alloy coating during plastic deformation was systematically revealed.Therefore, the study of the mechanism behind cavitation water jet-enhanced laser cladding of AlCoFeNi coatings is of significant importance in advancing the application of cavitation water jet peening technology.It has been marked in red font.

Question 2:Experimental Procedures: There are many details of the different work performed in this section, but it was not clear what the performed experiment was. By reading the other sections, I am assuming the order of procedures was: 1) laser cladding of the high entropy powder particles on the stainless steel substrate, 2) polishing, 3) CWJP for different times (2, 4, or 6 hours), and 4) hardness and microstructure testing to identify differences between the different processed samples. Is this correct? Please revisit this section to clarify the details such as what was different between the samples and how many replicates were performed for each treatment.

Reply: Thanks for asking. The experimental procedure is first laser cladding high-entropy alloy powder on stainless steel substrate, then polishing it, and then CWJP treatment for 2,4,6 hours. Finally, the hardness and microstructure of the specimens treated by CWJP at different times are tested, and the changes have been observed.

Question 3:Mechanical properties: The authors discuss how the proposed method has a positive impact on mechanical properties (including wear, fatigue, and corrosion resistance- see line 99) of high entropy alloys, but the only property measured in this study is the microhardness. The results would be significantly enhanced if another mechanical property could be measured as well.

 Reply: Thanks for the suggestion. The purpose of this paper is to study the influence of different cavitation water jet impact time on the microstructure and properties of high entropy alloy coating. Therefore, this paper studies the analysis of microstructure and grain distribution. Microstructure has a profound influence on its mechanical properties. For example, the smaller the grain size, the longer the grain boundary length, the greater the grain interface energy, the greater the tensile strength, yield strength and hardness of metal materials. However, the opinions provided by the experts are indeed worth considering. In the later stage, the relationship between mechanical properties and microstructure will be further studied.

Question 4:Microhardness: There is a significant decrease in hardness between points at 780 μm deep and 980 μm deep, according to Figure 6. Were any depths between those points measured? It would be interesting to see a more exact point of where the hardened layer ends, which could potentially be identified with more tests in that crucial area, rather than leaving a section of 200 um between measurements. Also, the authors could consider adding error bars to the measurements in Fig. 6.

Reply: Thanks for the suggestion. The hardness decreases significantly between the points of 780μm depth and 980μm depth, because the hardness of the substrate is significantly lower than that of the cladding layer. In this experiment, the effect of cavitation water jet on the surface layer of high entropy alloy coating was studied, so the microhardness of the substrate was deleted. According to the expert 's suggestion, figure 6 has been modified and error bars added.

Question 5:Table 4 lists the max, min, and mean values of grain size. Adding a standard deviation or other metric of dispersion of the measurements would help support the claim on line 274 that “the original specimen exhibits non-uniform grain sizes.”

Reply: Thanks for the suggestion. According to the expert 's suggestion, the standard deviation has been added in Table 4. 

Table 4 Statistical table of average grain size of specimens before and after CWJP treatment.

Grain size

Max/μm

Min/μm

Mean/μm

unCWJP

436.42

10.203

85.737±67.747

4hCWJP

310.69

13.203

67.044±48.106

Minor comments:

Question 6: Abstract: The authors state on line 11 of the abstract that an impact time of 4 hours results in “better” microhardness than 2 or 6 hours. By “better,” do the authors mean “harder?”

Reply: Thanks for asking. “better,” do mean “harder”. It can be seen from Fig.6 that when the CWJP impact time is 2 hours, the highest microhardness of the sample surface is 556.04 HV.When the CWJP impact time is 4 hours, the highest microhardness of the sample surface reaches 582.66 HV.When the CWJP impact time is 6 hours, the highest microhardness of the sample surface is 560.12 HV. 

Question 7: Figure 2: The text on lines 113-114 say Fig. 2 will be a schematic diagram, but the actual figure is a photograph of lab equipment, without any labels or information that informs how the process works. An actual schematic diagram similar to Fig. 3, would be helpful to understand the laser cladding process.

Reply: Thanks for asking. According to the expert 's suggestion, Fig.2 has been modified to inform the working principle of the process.

Question 8:Lines 187-188 describes a “shot-peened specimen without Cavitation Water Jet Peening (CWJP).” How can the specimen be shot-peened without the CWJP treatment? Do the authors mean a laser-clad and polished sample without CWJP? Or did they perform conventional shot-peening with some other method? If so, please describe that method.

Reply: Thanks for asking. Line 187-188 describes the 'shot peening specimen without Cavitation Water Jet Peening CWJP', which refers to the specimen treated by laser cladding and polishing, but not treated by cavitation water jet. The purpose of this paper is to study the effect of CWJP on its microstructure and mechanical properties under different impact time. Therefore, the surface morphology was observed under non-impact, 2h, 4h and 6h impact time.

Question 9:The font size used in the axes and colorbars for Figures 5, 8, and 9 are too small to read. Please increase the font size. Also, Figure 6 does not have any axes labels or units. Please correct.

Reply: Thanks for asking. According to the expert 's suggestion, The fonts in Figure 5, Figure 8 and Figure 9 have been increased, and the axis label has been added to Figure 6.

Question 10: On line 306, the authors state that “the untreated high-entropy alloy exhibits elongated equiaxed grains…” How can a grain be both elongated and equiaxed, since the two descriptions are opposites?

Reply: Thanks for asking. According to the expert 's suggestion, the article has been modified. Figure 10 depicts the Electron Backscatter Diffraction (EBSD) grain orientation maps for the high-entropy alloy specimens untreated and treated with 4 hours of CWJP. Fig.10a shows the grains of high entropy alloy without CWJP treatment. The largest grains have a size of approximately 400 μm, while the smallest are around 20 μm.Combined with the inverse pole figure, the orientation distribution law can be seen intuitively. It can be seen that the orientation distribution of the selected area of the material is mainly in the ( 001 ) direction and ( 111 ) direction. It has been marked in red font.

Question 11:Conclusion 2: On line 353, the authors state that the thickness of the hardened layer is approximately 650 um, but on line 231 and Fig. 6 of the text, the hardened layer thickness is given as 780 um. Please clarify.

Reply: Thanks for asking. According to the expert 's suggestion, 231 lines of  data have been modified. After CWJP strengthening, an effective hardened layer can be formed on the surface of the AlCoCrFeNi high-entropy alloy. The CWJP impact time has minimal influence on the thickness of the hardened layer, which is approximately 780 μm. When the CWJP impact time is 4 hours, the microhardness of the specimen surface surpasses that of 2 hours and 6 hours. This is attributed to insufficient impact intensity with shorter peening times, while excessively long peening times lead to erosion of the initially formed strengthened layer on the material surface. It has been marked in red font.

Reviewer 3 Report

Comments and Suggestions for Authors

The authors showed that high-entropy alloy was deposit onto substrate and than treated by Cavitation Water Jet Peening. The article is interesting and it provides some new data which will be useful for the application of this method to the surface treatment. Neverthelass the next comments should be considered before it can be accepted for publication:

line 108: "The powder particle size was 86.34 μm" I think the authors mean average particle size. Nevertheless it is better to do not show so many significant numbers after comma, and add the particle size distribution numbers. 

Table 2 How Chemical composition of high-entropy alloy was measured and it chenched after laser cladding?

Figure 4 On the diagram of nozzle structure (right hand image ), it is also will be good supply it with labels of what is what.

2.2. Microhardness measurement

It is necessary to add information about the load used for microhardness measurements.

3.2. Microhardness analysis

There are no axis labels in Figure 6. Each measurement should usually be accompanied by a measurement error. Plots sagging at X=1000 is also incorrect and should be fixed.

2.4. EBSD detection

What phase did you use for EBSD pattern indexing and why?

Figure 8. The axis labels are non visible. It is hard to understand whether the description in the text correspond to the EBSD measurements. The figure should be corrected than it will be possible to understand whether the authors conclusions are correct.

The same for Figure 9. The axis labels are non visible.

Table 4 provides the statistical data for the average grain size of specimens. To many significant numbers for such type of measurment. The error of grain size measurment or grains size variations should assesed and added to the table.

Figure 10.

The correspondence between the location of the sample and the given picture is unclear. The scale bars are absent on the image.

"reverse polarity map" should be Inverse pole figure (IPF) coloring. Should be corrected both in the caption and in the text.

Line 308 The orientation distribution is predominantly concentrated along the (001) direction, with a smaller fraction oriented in the (111) direction.  

This text should be supported by IPF of each EBSD map. I can not understand which color dominates on the image. I think all the colors can be seen equally, the only way I can agree with the text if inverse pole figures will be provided. 

Figure 11. No axis labels.

The article seriously lacks SEM (BSE) images of the microstructure of a cross-section of the coating before and after treatment.

Concluding the review, I think the paper is interesting itself, but it is made negligently. Thus before assesing the main conclusions of the paper, the figures should be corrected and some ammendements should be done. 

Author Response

Question1: line 108: "The powder particle size was 86.34 μm" I think the authors mean average particle size. Nevertheless it is better to do not show so many significant numbers after comma, and add the particle size distribution numbers.

Reply: Thanks for the suggestion. According to the expert 's suggestion, the line 108: powder particle size of 86.34μm ' refers to the average particle size of 86.34 microns, which has been modified in this paper. This study employed laser cladding technology to prepare materials, using 304 stainless steel as the substrate, with its chemical composition detailed in Table 1. It has been modified and marked in red.

Question2: Table 2 How Chemical composition of high-entropy alloy was measured and it chenched after laser cladding?

Reply: Thanks for asking. Table 2 is the chemical composition of high-entropy alloy powder, which is not the high-entropy alloy powder after laser cladding. This study employed laser cladding technology to prepare materials, using 304 stainless steel as the substrate, with its chemical composition detailed in Table 1. The cladding powder consisted of AlCoCrFeNi high-entropy alloy powder, with specific chemical composition provided in Table 2. The original text has been modified and marked in red.

Question3: Figure 4 On the diagram of nozzle structure (right hand image ), it is also will be good supply it with labels of what is what.

Reply: Thanks for the suggestion. According to the expert 's suggestion, Figure 4 has been modified and added the relevant labels.

2.2. Microhardness measurement

Question4: It is necessary to add information about the load used for microhardness measurements.

Reply: Thanks for the suggestion. According to the expert 's suggestion, the microhardness testing was carried out under a load of 0.981 N and a duration of 15 s to obtain the specimen hardness distribution profiles. It has been added to the text and marked in red. 

Question5: There are no axis labels in Figure 6. Each measurement should usually be accompanied by a measurement error. Plots sagging at X=1000 is also incorrect and should be fixed.

Reply: Thanks for the suggestion. According to the expert 's suggestion, the axis label has been added to Figure 6 and the measurement error has been added. The plots sagging at X = 1000 is also repaired.

2.4. EBSD detection

Question6: What phase did you use for EBSD pattern indexing and why?

Reply: Thanks for asking. In this paper, Fe-bcc phase is used for EBSD pattern indexing. According to the article 'Research on Microstructure and Properties of AlxCoCrFeNi High-Entropy Alloy.', it is found that the crystal structure of AlCoCrFeNi high-entropy alloy is mainly Fe-bcc phase. Therefore, the calibration of Fe-bcc phase in this paper can improve the accuracy of this experiment to a certain extent.

Question7:  Figure 8. The axis labels are non visible. It is hard to understand whether the description in the text correspond to the EBSD measurements. The figure should be corrected than it will be possible to understand whether the authors conclusions are correct.

Reply: Thanks for the suggestion. According to the expert 's suggestion, The axis label in Figure 8 has been corrected.

The same for Figure 9. The axis labels are non visible.

Reply: Thanks for the suggestion. According to the expert 's suggestion, the shaft label has been modified in Figure 9.

Question8: Table 4 provides the statistical data for the average grain size of specimens. To many significant numbers for such type of measurment. The error of grain size measurment or grains size variations should assesed and added to the table.

Reply: Thanks for the suggestion. According to the expert 's suggestion, The error of grain size measurement has been added to the statistical data of the average grain size of the sample provided in Table 4.

Table 4 Statistical table of average grain size of specimens before and after CWJP treatment.

Grain size

Max/μm

Min/μm

Mean/μm

unCWJP

436.42

10.203

85.737±67.747

4hCWJP

310.69

13.203

67.044±48.106

Question9: Figure 10.

The correspondence between the location of the sample and the given picture is unclear. The scale bars are absent on the image.

"reverse polarity map" should be Inverse pole figure (IPF) coloring. Should be corrected both in the caption and in the text.

Reply: Thanks for the suggestion. According to the expert 's suggestion, the scale has been added to Fig.10,and correct the antipole diagram in the caption and in the text.

Question10: Line 308 The orientation distribution is predominantly concentrated along the (001) direction, with a smaller fraction oriented in the (111) direction. 

This text should be supported by IPF of each EBSD map. I can not understand which color dominates on the image. I think all the colors can be seen equally, the only way I can agree with the text if inverse pole figures will be provided.

Reply: Thanks for the suggestion. According to the expert 's suggestion, the content of the text has been modified. Fig.10a shows the grains of high entropy alloy without CWJP treatment. The largest grains have a size of approximately 400 μm, while the smallest are around 20 μm.Combined with the inverse pole figure, the orientation distribution law can be seen intuitively. It can be seen that the orientation distribution of the selected area of the material is mainly in the ( 001 ) direction and ( 111 ) direction. It has been marked in red font. And the inverse pole figures has been re-inserted.

Figure 11. No axis labels.

Reply: Thanks for the suggestion. According to the expert 's suggestion, The axis label has been added in Fig.11.

The article seriously lacks SEM (BSE) images of the microstructure of a cross-section of the coating before and after treatment.

Reply: Thanks for the suggestion.The purpose of this paper is to study the effect of different cavitation water jet impact time on the microstructure and properties of high-entropy alloy coatings. Therefore, this paper studies the analysis of the microstructure and grain distribution of the cavitation water jet impact surface. However, the opinions provided by experts are indeed worth considering, and the test will be further optimized in the later stage.

Round 2

Reviewer 2 Report

Comments and Suggestions for Authors

The authors addressed many of my concerns and made many positive changes to the manuscript. However, some of their explanations and changes only appear in the response letter, and the problematic or confusing language remains in the manuscript. I will recommend this manuscript for publication once the authors make those changes in the actual manuscript text. See my comments below.

1) Experimental Procedures: The authors clarified the order of experiments in the response letter, but the text on page 6 of the manuscript appears unchanged. Please add a quick clarification to the experimental order. Also, the number of produced samples should be explicitly stated.

2) Abstract: The authors clarified in the response letter that “better” does indeed mean “harder.” However, the manuscript text still says “better,” which is an ambiguous term. The authors should change the wording in the manuscript.

3) On page 7, the description for figure 5a includes a “shot-peened specimen without Cavitation Water Jet Peening (CWJP).” This is a confusing description, since that specimen is not shot-peened at all, but rather laser clad and polished. The authors clarified their meaning in the response letter, but left the confusing description in the manuscript text. Please correct in the manuscript text.

4) The font size used in the axes for Figures 8 and 9 are still too small to read. Please significantly increase the font size. Also, why do the histogram bins in Figure 8 have different widths? For a fair visual comparison, both Fig. 8a and 8b should have the same exact bin size. Please correct.

5) On page 13, in the description of Figure 4b, the authors state that “After 4 hours of CWJP treatment, the high-entropy alloy undergoes dynamic recrystallization in the impact region, resulting in elongated and equiaxed grains.” While slightly different from the first iteration, this description is still confusing since a single grain cannot be both elongated and equiaxed. Do the authors mean that the dynamic recrystallization caused some grains to be equiaxed and some other grains to be elongated? If so, is there some physical explanation for this- why would the dynamic recrystallization not cause all grains to be equiaxed? Please clarify in the manuscript text.

Author Response

Question1: Experimental Procedures: The authors clarified the order of experiments in the response letter, but the text on page 6 of the manuscript appears unchanged. Please add a quick clarification to the experimental order. Also, the number of produced samples should be explicitly stated.

Reply: Thanks for the suggestion. The experimental sequence has been clarified on page 6 of the manuscript, along with a description of the number of samples. In this study, the experimental procedure is first laser cladding high-entropy alloy powder on stainless steel substrate, and a total of 4 specimens are prepared by laser cladding technology, then polishing it, and then CWJP treatment for 2,4,6 hours. Before the impact test, the specimen was cleaned in an ultrasonic cleaner for 20 minutes, followed by drying. It has been marked in blue font.

Question2: Abstract: The authors clarified in the response letter that “better” does indeed mean “harder.” However, the manuscript text still says “better,” which is an ambiguous term. The authors should change the wording in the manuscript.

Reply: Thanks for the suggestion. The “better” in the abstract has been changed to “harder”. The results show that an effective hardening layer can be formed on the surface layer of AlCoCrFeNi high-entropy alloy after CWJP. When the CWJP impact time is 4 hours, the microhardness of the surface layer of the specimen is harder than that of 2 hours and 6 hours, and the CWJP impact time has little effect on the thickness of the hardening layer. It has been marked in blue font.

Question3: On page 7, the description for figure 5a includes a “shot-peened specimen without Cavitation Water Jet Peening (CWJP).” This is a confusing description, since that specimen is not shot-peened at all, but rather laser clad and polished. The authors clarified their meaning in the response letter, but left the confusing description in the manuscript text. Please correct in the manuscript text.

Reply: Thanks for the suggestion. In the manuscript text, the shot peening sample ' without cavitation water shot peening ( CWJP ) ' in Figure 5a has been modified. Figure 5a illustrates shows the surface morphology of the specimens after laser cladding and polishing, but without treated by cavitation water jet peening ( CWJP ). It has been marked in blue font.

Question4: The font size used in the axes for Figures 8 and 9 are still too small to read. Please significantly increase the font size. Also, why do the histogram bins in Figure 8 have different widths? For a fair visual comparison, both Fig. 8a and 8b should have the same exact bin size. Please correct.

Reply: Thanks for the suggestion. The font size in Figure 8 and 9 has been significantly increased, and the histogram in Figure 8 has been modified.

Question5: On page 13, in the description of Figure 4b, the authors state that “After 4 hours of CWJP treatment, the high-entropy alloy undergoes dynamic recrystallization in the impact region, resulting in elongated and equiaxed grains.” While slightly different from the first iteration, this description is still confusing since a single grain cannot be both elongated and equiaxed. Do the authors mean that the dynamic recrystallization caused some grains to be equiaxed and some other grains to be elongated? If so, is there some physical explanation for this- why would the dynamic recrystallization not cause all grains to be equiaxed? Please clarify in the manuscript text.

Reply: Thanks for the suggestion. After CWJP treatment, the shock wave deforms the metal. Due to the increase of atomic diffusion capacity, the elongated and broken grains become new uniform and fine equiaxed grains by re-nucleation and growth. It has been modified in the text of the manuscript. It has been marked in blue font.

Reviewer 3 Report

Comments and Suggestions for Authors

The average grain size should be given in accordance with the uncertainty in measured value. The significant numbers in errors usually should not exceed two digits. This point should be corrected for HV value (3.2 Microhardness analysis), for Table 4 and text below.

Figure 10 (a) and (b), the scale bar value should be enlarged, in current state it is nonvisible. Figure 10 (c) is not Inverse pole figure itself, it is standard stereographic triangle which depict colouring the grains in accordence to its orientation. The caption to the figure should be corrected. I also insist on adding pole figures for (a) and (b) to evaluate the preferential orientation of the grains in the sample. Preferable orientation is mentioned in the text below the figure 10.

Author Response

Question1: The average grain size should be given in accordance with the uncertainty in measured value. The significant numbers in errors usually should not exceed two digits. This point should be corrected for HV value (3.2 Microhardness analysis), for Table 4 and text below.

Reply: Thanks for the suggestion. The significant numbers has been modified.

Table 4 Statistical table of average grain size of specimens before and after CWJP treatment.

Grain size

Max/μm

Min/μm

Mean/μm

unCWJP

436.42

10.20

85.73±67.74

4hCWJP

310.69

13.20

67.04±48.10

Question2: Figure 10 (a) and (b), the scale bar value should be enlarged, in current state it is nonvisible. Figure 10 (c) is not Inverse pole figure itself, it is standard stereographic triangle which depict colouring the grains in accordence to its orientation. The caption to the figure should be corrected. I also insist on adding pole figures for (a) and (b) to evaluate the preferential orientation of the grains in the sample. Preferable orientation is mentioned in the text below the figure 10.

Reply: Thanks for the suggestion. The scale bars in Fig. 10 ( a ) and ( b ) have been magnified and a pole figure has been added to the manuscript. Fig.10a shows the grains of high entropy alloy without CWJP treatment. The largest grains have a size of approximately 400 μm, while the smallest are around 20 μm. Figure 11 depicts the pole figures of high-entropy alloy before and after CWJP treatment. Combined with the inverse pole figure, the orientation distribution law can be seen intuitively. It can be seen that the orientation distribution of the selected area of the material is mainly in the ( 001 ) direction and ( 111 ) direction. It has been marked in blue font.
